# *HsGA20ox1*, *HsGA3ox1*, and *HsGA2ox1* Are Involved in Endogenous Gibberellin Regulation Within *Heracleum sosnowskyi* Ovaries After Gibberellin A_3_ Treatment

**DOI:** 10.3390/ijms26104480

**Published:** 2025-05-08

**Authors:** Tautvydas Žalnierius, Dominykas Laibakojis, Saulė Rapalytė, Jurga Būdienė, Sigita Jurkonienė

**Affiliations:** 1Laboratory of Plant Physiology, Nature Research Centre, Akademijos Str. 2, 08412 Vilnius, Lithuania; dominykas.laibakojis@gamtc.lt (D.L.); saule.rapalyte@gamtc.lt (S.R.); sigita.jurkoniene@gamtc.lt (S.J.); 2Laboratory of Chemical and Behavior Ecology, Nature Research Centre, Akademijos Str. 2, 08412 Vilnius, Lithuania; jurga.budiene@gamtc.lt

**Keywords:** Sosnowsky’s hogweed, Apiaceae, invasive plant, gibberellic acid, genome-wide analysis, early fruit development, 2-oxoglutarate-dependent dioxygenases

## Abstract

This study aims to investigate the endogenous gibberellin levels and related genes analysis of noxious invasive weed *Heracleum sosnowskyi*. Genome-wide identification, phylogenetic analysis, conserved motif analysis, and gene structure characterization of GA-oxidases were performed. We analysed endogenous GAs levels and the expression of target *HsGAoxs* in response to GA_3_ within *H. sosnowskyi* developing ovaries. Twenty-seven *HsGAoxs* genes were identified, distributed across eleven chromosomes. Phylogenetic analysis classified proteins into the HsGA20ox, C_19_-HsGA2ox, and HsGA3ox subfamilies, facilitating functional predictions. Among the thirteen HsGA2ox protein members, there were no C_20_-GA2ox subfamily that distinguish *H. sosnowskyi* from other model plant species. The analysis of gene structure and conserved motifs confirmed the phylogenetic grouping and suggested that the evolutionary pattern was maintained within these subfamilies. The observed increase in precursor and bioactive GA levels provides evidence that they play a crucial role in promoting fruit growth. Ovary phenotypes reflected the timing of peak gibberellin levels, specifically during the cell expansion period. Exogenous GA_3_ treatment promoted *HsGA3ox1* expression within both the central and lateral regions of the umbel ovaries. Overall, the results show that GA levels are precisely regulated by multiple *HsGAox* genes for stable early fruit development, and that disturbances in this stability affect fruit development. This opens up the possibility of investigating the role of GA in *H. sosnowskyi* fruit formation and developing measures for invasion control.

## 1. Introduction

Gibberellins (GAs) are a large group of plant hormones, encompassing 136 chemically similar compounds found in plants, fungi, and bacteria [1,2]. They belong to diterpene carboxylic acids with a tetracyclic *ent*-gibberellane skeleton, composed of 19–20 carbon atoms [3]. Biologically active GAs act as endogenous plant growth and development regulators, transmitting internal and environmental signals within the organism [1]. They stimulate cell elongation, cell division, regulate seed germination, control the transition between juvenile and generative plant phases, induce flowering, influence sex determination, and play a role in fruit setting and growth processes [4,5]. However, only a small group are biologically active: gibberellin A_1_ (GA_1_), gibberellin A_3_ (GA_3_), gibberellin A_4_ (GA_4_), and gibberellin A_7_ (GA_7_) [1]. Interestingly, GA_1_ and GA_4_ are the dominant forms in plant tissues [6,7,8]. GA_3_ and GA_7_ metabolism in plants is not a common process, and these GA species are present in trace amounts; however, microorganisms can metabolize GA_20_ to GA_3_, or GA_7_ directly from GA_4_, involving GA desaturases [9,10]. For plants, it is crucial to regulate GA levels precisely; however, the chemical structure specificity of 1,2-unsaturated GAs (GA_3_ and GA_7_) obstructs their inactivation in plants by 2β-hydroxylation, which results in their prolonged activity and may explain their low amounts in plants [2,10].

GA biosynthesis has been studied in many plant species, and it occurs in three stages according to subcellular localization, where involved enzymes act [4,11]. The first stage of GA biosynthesis takes place in plastids, where geranylgeranyl diphosphate is converted to *ent*-kaurene through an intermediate, catalysed by terpene synthases: *ent*-copalyl diphosphate synthase (CPS) and *ent*-kaurene synthase (KS) [12,13]. Subsequently, in the outer plastid membrane and endoplasmic reticulum, cytochrome P450 monooxygenases catalyse oxidation reactions, during which *ent*-kaurene is converted through four intermediates to GA_12_ [14,15]. The third stage of synthesis occurs in the cell cytoplasm, where the common precursor GA_12_ is oxidised by two distinct soluble 2-oxoglutarate-dependent dioxygenases (2-ODDs)—GA 20-oxidase (GA20ox) and GA 3-oxidase (GA3ox)—leading to the biologically active GA_1_ or GA_4_ [11,16,17]. To keep optimal levels of active GAs species within tissues, the inactivation process is essential for plant growth and development. The main mechanism involves two groups of 2-ODDs enzymes, differentiated by their substrate specificity—C_19_-GA 2-oxidase and C_20_-GA 2-oxidase—both of which utilise 2β-hydroxylation to catabolize precursors and biologically active GAs into inactive products [2,18,19,20].

The 2-ODD superfamily contains four enzyme subfamilies (GA20ox, GA3ox, C_19_-GA2ox and C_20_-GA2ox), involved in GA biosynthesis and catabolism, which are encoded by small multigene families and are conserved between multiple species [19,21,22,23]. Genes of GAoxs subfamilies were first isolated in model plant species: *GA20ox* in pumpkin [24], *GA3ox* in *Arabidopsis thaliana* [25], and *GA2ox* in runner bean (*Phaseolus coccineus*) and in *Arabidopsis thaliana* [26]. To date, GAoxs genes have been identified in many commercially important plants, such as cucumber [18], watermelon [27], tomato [28,29,30], maize [31], rice [3,32], peach [33], grapevine [34], wild cherry, wild strawberry [35], breadfruit [36], and others. With the identification of GAoxs enzymes and their genes, the mechanism of fruit set became better understood, which opened opportunities to manipulate fruit set and induce parthenocarpy [28,29,37,38].

*Heracleum sosnowskyi* is a noxious invasive weed species enlisted into Invasive Alien Species of Union concern and national lists of invasive species in many EU countries [39,40]. *H. sosnowskyi* propagates only by seeds and after bearing them, eventually dies [41,42]. Interestingly, *Pastinaca sativa* (Apiaceae) is adapted to cope with some pests by producing seedless fruits. This strategy enables it to regulate outbreaks of herbivores [43]. Previous studies have indicated an exogenous GA_3_ effect in inducing seedlessness in *Heracleum sosnowskyi* [44,45]. However, to date, analysis of endogenous GAs and related genes has not been available. With the recent article on the *H. sosnowskyi* genome, GAoxs and gene analysis have become accessible in this invasive species [46].

In this study, we identified 27 putative GAoxs enzymes in the *H. sosnowskyi* genome. Phylogenetic analysis clustered these proteins into three subgroups: GA20ox, GA3ox, and C_19_-GA2ox. We identified the effects of exogenous GA_3_ on the expression of *HsGA20ox1*, *HsGA3ox1*, and *HsGA2ox1*, as well as on the endogenous GA profiles in gradually opening *H. sosnowskyi* flowers. These results open opportunities to further analyse the role of GAs in *H. sosnowskyi* fruit set mechanism and develop invasion control strategies.

## 2. Results

### 2.1. Genome-Wide Identification and Analysis of GA-Oxidase Genes in Heracleum sosnowskyi

To identify *GAox* genes from *GA20ox*, *GA2ox*, and *GA3ox* subfamilies in *Heracleum sosnowskyi*, the amino acid sequences of GA20ox, GA2ox, and GA3ox from *Arabidopsis thaliana* and *Oryza sativa* were used as a reference in BLASTp program against the *Heracleum sosnowskyi* genome. A total of 32 putative protein homologues were obtained. However, five protein candidates did not possess 2OG-Fe(II) oxygenase (PF03171) or non-haem dioxygenase N-terminal (PF14226) domains, as confirmed by InterPro Pfam (available online: https://www.ebi.ac.uk/interpro accessed on 10 March 2025) (Appendix A). Thus, we concluded that *H. sosnowskyi* contains 27 GAox family members. We identified and renamed *GA20ox*, *GA2ox*, and *GA3ox* genes in the *H. sosnowskyi* genome: nine *HsGA20ox1–9*, thirteen *HsGA2ox1–13*, and five *HsGA3ox1–5* (Table 1).

The CDS lengths of the *HsGA20ox*, *HsGA2ox*, and *HsGA3ox* genes varied from 978 (*HsGA20ox7*) to 1221 nt (*HsGA20ox4*), from 945 (*HsGA2ox8*) to 1665 nt (*HsGA2ox2*), and from 1020 (*HsGA3ox5*) to 1068 nt (*HsGA3ox2*), respectively. In HsGAox subfamilies, amino acid residue ranges slightly differed: HsGA20ox (325–406 aa), HsGA2ox (314–554 aa), and HsGA3ox (339–355 aa). The theoretical molecular weight ranged from 36.81 to 46.14 kDa (HsGA20ox), for HsGA2ox from 35.39 to 61.92 kDa, and for HsGA3ox from 38.00 to 40.02 kDa (Table 1). Using Plant-mPLoc software version 2.0, we identified the subcellular localization of the HsGAox proteins. Analysis revealed that all members from HsGA20ox, HsGA2ox, and HsGA3ox are localized in the cytoplasm. Interestingly, only one member, HsGA2ox2 protein, displayed localization in both the cytoplasm and nucleus (Table 1).

To determine the exact genomic positions of *HsGAox* genes, chromosome localization was carried out. A total of 27 *HsGAox* genes were widely distributed across eleven *H. sosnowskyi* chromosomes (Figure 1). *HsGA20ox* gene subfamily members were identified to be distributed in chromosomes 1, 2, 3, 7, and 8. The vast majority of gene loci are localized in the lower arm of the chromosomes; however, *HsGA20ox7* and *HsGA20ox8* are situated in the upper arm and in the middle of chromosome 3, respectively. Thirteen *HsGA2ox* genes are scattered across six chromosomes. Eight *HsGA2ox1*, *2*, *5*, *6*, *9*, *10*, *11*, and *12* loci are located in the lower arm of the chromosomes 2, 4, 5, 6, 10, and 11. The rest of the genes localized in the upper arm of the chromosomes 6, 10, and 11. The *H. sosnowskyi GA3ox* subfamily genes are scattered among chromosomes 5, 9, 10, and 11. *HsGA3ox1, 4*, and *5* are located in the lower arm of chromosomes 11 and 5, respectively. The *HsGA3ox3* locus was located at the end of the upper arm of chromosome 9. Interestingly, chromosomes 1–6 are considered longer than others, but *HsGAox* genes are distributed across all chromosomes and do not show a dominant correlation to chromosome length.

### 2.2. Phylogenetic Analysis of the HsGAox Genes

To explore the phylogenetic relationship of HsGAox with homologues from other taxa, we involved two model species, *Arabidopsis thaliana* and *Oryza sativa*, in the analysis. A maximum-likelihood (ML) phylogenetic tree was built based on alignments of the complete protein sequences from 27 *Heracleum sosnowskyi*, 16 *Arabidopsis thaliana*, and 21 *Oryza sativa* found in NCBI and Phytozome databases. Four subfamilies (GA20ox, C_19_-GA2ox, C_20_-GA2ox, and GA3ox) were identified. However, *Heracleum sosnowskyi* proteins clustered into three different subfamilies: GA20ox, C_19_-GA2ox, and GA3ox. Interestingly, among thirteen HsGA2ox members, none belonged to the C_20_-GA2ox subfamily. Moreover, HsGA20ox1 and -2, HsGA20ox5 and -6 appear to be duplicate proteins. These results allow us to make speculations on the functions of the HsGA20ox, HsGA2ox, and HsGA3ox according to the phylogenetic classification mentioned above (Figure 2).

### 2.3. Conserved Motif and Gene Structure Analysis of the HsGAox Genes

To support the phylogenetic analysis, we predicted conserved motifs of the HsGAox protein family using the Multiple Em for Motif Elicitation (MEME) online tool version 5.7.7. Ten conserved motifs for HsGA20ox, C_19_-HsGA2ox, and HsGA3ox subfamily proteins were obtained and ranged from 21 to 50 amino acid residues (Figure 3B). The assessment of motifs 1–10 with InterPro Pfam revealed that motif 1 exhibits 2OG-Fe(II) oxygenase and motif 2 exhibits non-haem dioxygenase N-terminal domains. Except for HsGA3ox, all HsGA20oxs and HsGA2oxs were endowed with eight motifs in the same pattern. In contrast, HsGA3oxs contained seven motifs (HsGA3ox4 had six motifs); moreover, motif composition of the HsGA3oxs was distinct from HsGA20oxs and HsGA2oxs by the C-terminal end, where motif 6 or motif 8 was absent. In more detail, HsGA20oxs subfamily proteins are distinct from other HsGAoxs, containing a family-specific motif 6 at N-terminal end of the sequence. Furthermore, motif 10 is unique to HsGA20ox and HsGA3ox subfamily members; in contrast, in the same location of HsGA2ox sequences, motif 5 is observed.

We analyzed the intron and exon compositions, as well as dispositions in the gene structure of *HsGAoxs* (Figure 3C). Analysis revealed that the majority of *HsGA20ox* members had a consistent number of three exons and two introns, whereas *HsGA20ox4* and -*7* number of exons varied from four to five, respectively. *HsGA2ox* subfamily genes contained three to five exons: 10 members (*HsGA2ox1*, *3*, *5*–*12*) had three exons, *HsGA2ox4*—four exons, and *HsGA2ox2*,*13*—five exons. The *HsGA3ox* subfamily (*HsGA3ox1*–*5*) showed a relatively simple gene structure, consisting of two exons and one intron. In summary, the length of introns varied within three *HsGAoxs* gene subfamilies, contributing to the overall size differences among gene members in these subfamilies. To sum up, gene structure and motif analyses support the phylogenetic findings, indicating the conserved evolution of these gene subfamilies in all three species.

### 2.4. Analysis of HsGAoxs Expression in Response to GA_3_ Within Heracleum sosnowskyi Developing Ovaries

To evaluate the exogenous GA_3_ effect on the expression of GA biosynthesis enzyme genes, the real-time quantitative PCR (qPCR) method was used to determine the quantity of gene transcripts in the terminal umbel ovaries, which were collected from the central and lateral parts at different stages of development (Figure 4A). Specific primers for the putative *HsGA20ox1*, *HsGA3ox1*, and *HsGA2ox1* genes were designed and used in the analysis. It was found that the *HsGA3ox1* gene was most intensely expressed at 3 and 10 days after the start of the experiment, both in the central and lateral parts of the terminal umbel. The quantity of *HsGA20ox1* transcripts in the terminal umbel ovaries peaked 3 days after the start of the experiment, but the expression of this gene decreased after 10 days from the onset of flower spreading (Figure 4B). Due to the effect of GA_3_, the relative abundance of *HsGA20ox1* gene transcripts decreased by almost half on the third day of the experiment, reaching 8.9% in the central part (*p* = 0.04) and 8.7% in the lateral part (*p* = 0.85). After 10 days, the expression of this gene was even weaker in both the central (2.4%) and lateral (2.2%) parts. The analysis of *HsGA3ox1* gene expression revealed that 3 days after the GA_3_ treatment, the expression of this gene decreased from 41.5% to 7.18% (*p* = 0.8) in the central part of the umbel and from 45.3% to 7.27% (*p* = 0.72) in the lateral part. Ten days after the GA_3_ treatment, the relative abundance of *HsGA3ox1* gene transcripts significantly increased, reaching 235% (*p* < 0.001) in the central part and 168% (*p* = 0.0019) in the lateral part of the umbel.

### 2.5. Changes in Endogenous GA Levels in Heracleum sosnowskyi Ovaries After Treatment with Exogenous GA_3_

The dynamics of endogenous GAs were also analyzed in another experiment in which we aimed to determine the distribution of metabolites of the C-13-hydroxylated GA biosynthesis pathway in the central and lateral parts of the terminal umbel after GA_3_ treatment (Figure 4C). The obtained results revealed that the level of biologically active GA_1_ was extremely low in both parts of the umbel at 0 and 3 days after application (DAA): 963 pg/g FW in the central part and 264 pg/g FW in the lateral part. However, at a later stage of ovary development (10 DAA), the hormone level in the central part of the umbel significantly increased to 16.3 ng/g FW (*p* = 0.011) and to 43.6 ng/g FW (*p* < 0.001) in the lateral part. Interestingly, the GA_1_ content in the ovaries of the central and lateral parts differed significantly, by almost a twofold difference (*p* < 0.001). The level of the precursor GA_20_ reached its peak in the central (72.47 ng/g FW) and lateral (46.64 ng/g FW) parts of the umbel during the 3 DAA period. However, during the 10 DAA period, the level of this metabolite significantly decreased to 4.37 ng/g FW (*p* < 0.001) and 2.87 ng/g FW (*p* = 0.005) in the ovaries of the central and lateral parts, respectively. The level of the GA_20_ catabolite (GA_29_) remained low throughout the experiment. The level of the GA_1_ catabolic form (GA_8_) increased in the central part (12.8 ng/g FW) and in the lateral part (7.3 ng/g FW) during the 3 DAA period, and then significantly decreased to 1.19 ng/g FW (*p* < 0.001) and 0.89 ng/g FW (*p* = 0.059) in each part of the umbel, respectively, during the later period (10 DAA). The highest levels of early hormone precursors GA_44_ and GA_19_ in the terminal ovaries were detected only at the 3 DAA developmental stage. Interestingly, in the early stage of ovary development, GA_44_ was found four times more abundant in the central part of ovary tissues (32.2 ng/g FW) than in the lateral part (7.35 ng/g FW), and the difference between them was significant (*p* < 0.001). After spraying the terminal umbels with exogenous GA_3_, the GA_1_ content increased insignificantly in the ovaries of the central part from 0.963 to 5.15 ng/g FW (*p* = 0.95), and significantly in the lateral part from 0.264 to 20.9 ng/g FW (*p* < 0.001) after 3 DAA. Ten days after application with GA_3_ solution, an uneven distribution of GA_1_ hormone levels was observed depending on the umbel area: significantly decreased in the central part to 388 pg/g FW (*p* < 0.001) and significantly decreased in the lateral part to 26.86 ng/g FW (*p* < 0.01). Due to GA_3_ treatment, the level of catabolite GA_29_ increased dramatically in the ovaries of the lateral part of the umbel: 6.66 ng/g FW (*p* < 0.001) and 6.28 ng/g FW (*p* < 0.01) in the 3 DAA and 10 DAA periods, respectively.

### 2.6. GA_3_ Impact on the Phenotype of Heracleum sosnowskyi Ovaries

Morphometric analysis of the samples revealed that the development of mericarps is not uniform in the central and lateral parts of the umbel: after 10 days, the mericarps in the central part of the umbel were 11% shorter (U = 2000.5, *p* < 0.001), 10% narrower (U = 2385, *p* < 0.01), and 9% lighter (U = 2185.5, *p* < 0.001) than the mericarps from the lateral part of the umbel (Figure 4D,E). Three days after GA_3_ 150 mg/L treatment, there was a statistically significant decrease in mericarp length (U = 1459, *p* < 0.001) and weight (U = 2004, *p* < 0.01) in the central part, and width (U = 2203.5, *p* < 0.001) in the lateral part. Ten days after application, there was a statistically significant decrease in mericarp length (21%) (U = 1656, *p* < 0.001), width (30%) (U = 1020, *p* < 0.001), and weight (59%) (U = 1008.5, *p* < 0.001) in the central part, and a decrease in length (20%) (U = 1337, *p* < 0.001), width (31%) (U = 712.5, *p* < 0.001), and weight (41%) (U = 703, *p* < 0.001) in the lateral part (Figure 4E).

## 3. Discussion

Gibberellins play an important role in the processes of fruit-set, development, and ripening [47,48]. In horticulture, plants are often sprayed with bioactive GAs to obtain high-quality seedless fruits, but this can have a negative impact on the histological and morphological structure of the fruit [49,50,51,52]. The 2-ODD protein superfamily is the second-largest enzyme family in plants. Moreover, 2-ODDs have a crucial role in leading oxygenation or hydroxylation in various plant metabolic events [19]. Members of 2-ODDs have been identified in many model plant species: *Arabidopsis thaliana*, rice, cucumber, tomato, and others [18,26,53,54,55,56,57,58]. However, information about 2-ODDs in invasive *Heracleum sosnowskyi* is very limited. A recent study by [46] made the genome of *H. sosnowskyi* available for genome-wide analysis. According to GAoxs protein sequences from *Arabidopsis thaliana* and *Oryza sativa* [59], we identified and named nine *HsGA20ox1–9*, thirteen *HsGA2ox1–13*, and five *HsGA3ox1–5* protein homologues from *Heracleum sosnowskyi* (Table 1). All putative GAoxs contained 2OG-Fe(II) oxygenase (PF03171) and non-haem dioxygenase N-terminal (PF14226) domains, which are known to be specific for 2-ODDs [60,61]. In our study, we predicted the 3D protein structures of several HsGAoxs and identified that both previously named domains are located in the reaction crevice of the proteins (Figure 3A). This aligns with the previous models of suggested GAoxs in rice [62]. Interestingly, HsGA20ox1 and HsGA20ox2 seem to be duplicates, which have one amino acid substitution caused by single nucleotide polymorphism (Figure 3A). Gene duplication is generally seen as a major driver of genes developing different functions, a process steered by natural selection, which is important to consider in further HsGAox analysis [63,64]. Phylogenetic analysis suggested that HsGA20oxs belong to three distinct clades; similarly, HsGA3oxs seem to belong to two different paraphyletic groups. Interestingly, we have not found any members of C_20_-GA2oxs (Figure 2). However, the functional characterization using C_20_- and C_19_-GA substrates of HsGA2oxs might improve our classification. Previous studies indicate that GA2oxs are composed of three groups: C_19_-GA2ox-I, C_19_-GA2ox-II, and C_20_-GA2ox-I [11,57,65,66]. In our study, we observed that *H. sosnowskyi* C_19_-2ox members cluster together with *Oryza sativa* and *Arabidopsis thaliana* members into two monophyletic groups: C_19_-GA2ox-I and C_19_-GA2ox-II (Figure 2). These findings suggest that C_19_-HsGA2oxs might have functional differences among the groups. Furthermore, it is known that different *GAox* genes display significant variations in their expression across different tissues [38,67,68,69,70]. A comparison of the phylogenetic tree with the gene structure and motif analysis reveals that the most closely related members within subfamilies share similar gene structures and motif compositions (Figure 2 and Figure 3). Interestingly, HsGA20ox and HsGA2ox are found in greater numbers than HsGA3oxs, similarly to *A. thaliana*, *Oryza sativa*, *Cucumis sativus*, and *Salix matsudana* [18,62,71,72]. To delve into the evolutionary relationships of GAox genes, we analyzed motif composition and gene structure (Figure 3). However, GA20ox and GA3ox subfamilies were distinct from GA2ox by the LPWKET motif, which is characteristic of GA20oxs and for some GA3oxs [38,62,71]. Although motif 9 was observed in all C_19_-HsGA2oxs, it is distinct from the rest of the 2-ODD subfamilies [62]. Gene structure revealed that all *HsGA20oxs* contain three exons and two introns, except for *HsGA20ox4* and *-7*, which are similar to the structure of the GA20ox subfamily in *Arabidopsis thaliana* [57]. *HsGA3oxs* have a simple gene structure of one intron and two exons, consistent with those in *A. thaliana*, *Cucumis sativus*, and *Oryza sativa* [57,59]. Our findings reveal that closely related members within GAox subfamilies possess similar structures and motifs, implying shared functions.

A more detailed analysis of terminal umbel ovaries in the early stages of fruit development revealed a differential distribution of GA and metabolites between the central and lateral regions. This was particularly reflected in a significant increase in the levels of metabolites GA_44_ and GA_20_ in the control samples of the central umbel region at 0 and 3 days after application (DAA) (Figure 4C). This inter-regional distribution may be due to the complex umbel architecture, as the flowers in the lateral part of the umbel open first [73]. In addition, the literature indicates that GA is involved in the sex determination of male flowers [74,75,76], with high levels of GA precursors and the hormone GA_4_ accumulating in the floral parts of stamens [77]. As is known, male flowers in the umbels of some Apiaceae family representatives, such as *Zizia aurea* and *Thaspium barbinode*, are distributed centripetally [78]. However, the terminal umbel of *Heracleum mantegazzianum* is usually composed only of hermaphrodite flowers [73]. There is no literature data on the distribution of hermaphrodite and male flowers in the terminal umbel of *H. sosnowskyi*, but based on our endogenous GA profiles (Figure 4C), it can be assumed that there should be more of them in the center of the *H. sosnowskyi* umbel. Three days after exogenous GA_3_ treatment, the accumulation of the metabolite GA_20_ was significantly reduced in the ovaries of both parts of the umbel, but the level of the hormone GA_1_ in the ovary tissues increased (Figure 4C). Ten days after application, the synthesis of the hormone GA_1_ in the ovary tissues of the central part of the umbel was completely suppressed, but only partially in the lateral part of the umbel. The inhibitory effect of exogenous GA_3_ on GA biosynthesis can be identified by a significant decrease in the precursor GA_20_ and a marked increase in the amount of the catabolite GA_29_ in the ovaries of both parts of the umbel. However, it remains unclear why exogenous GA_3_ forms a GA_1_ gradient in different parts of the umbel (Figure 4C). It is known that an increase in the level of biologically active GA forms in the early stages of fruit development leads to the development of parthenocarpic fruit set [37,48,50,79,80]. It should be noted that in this experiment, flower pollination was not controlled, so the GA_1_ content at 10 DAA could have increased due to natural fertilization (Figure 4C).

Before analyzing the abundance of GA biosynthesis gene (*HsGA20ox1*, *HsGA3ox1*, and *HsGA2ox1*) transcripts, these genes were cloned from *H. sosnowskyi* umbel tissues collected 10 days after application (DAA), and specific primers were designed for them. It was observed that in the early stage of fruit development (3 DAA), *HsGA20ox1* was more highly expressed, but its expression decreased in later stages (10 DAA) (Figure 4B). On the other hand, the expression of the metabolic *HsGA3ox1* gene was much more intense, with no differences detected between umbel regions. Other studies have indicated that the increase in *GA20ox* gene expression and the decrease in *GA2ox* are essential factors in regulating GA biosynthesis during fruit set [47,50,81,82]. It should also be noted that in other plant species (e.g., white clover, tomato, cultivated rice), more than one GA oxidase gene is involved in fruit set control [28,48,77,79,83]. However, our results suggest that intense fruit development is dependent on endogenous GA, which is supported by the obtained levels of endogenous GA (Figure 4C). Additionally, the results of phenotypic and morphometric analysis of ovaries (Figure 4D,E) correspond to the peak of gibberellin during the cell expansion phase observed in the model plant tomato [84,85]. However, the effect of exogenous GA_3_ stimulated *HsGA3ox1* expression in the central and lateral parts of the umbel ovaries, while the inhibition of *HsGA2ox1* gene expression, as speculated, was not detected (Figure 4B). Furthermore, the obtained expression profile of the *HsGA20ox1* gene at 10 DAA, both in the control and after GA_3_ treatment, suggests that fruit set had already occurred, as seen in some species such as pear, where after a similar treatment with exogenous biologically active GA, the expression of genes *PbGA20ox1*, *PbGA20ox2*, and *PbGA20ox3* is suppressed [38]. Our results align with the suppressed expression of paralogous genes of the PbGA20ox (*PbGA20ox1*, *PbGA20ox3*) in pollinated pear ovaries [38].

## 4. Materials and Methods

### 4.1. Research Object and Growth Conditions

An invasive habitat of *Heracleum sosnowskyi* was selected for field trials, and an experimental field was established. The test site was located in an anthropogenic *H. sosnowskyi* habitat with an area of 6.48 ha in an abandoned area in Vilnius, Lithuania (54.739749° N, 25.258871° E WGS80). The experiment field was fenced with a “stop” line and marked with signs informing about the plant research being conducted. The studies were conducted on *H. sosnowskyi* plants during the flowering and fruit formation stages from 2021. Samples were collected from 3 plants. The terminal umbel was divided into central and lateral parts following the guidelines [86].

### 4.2. Application of Plants with GA_3_

Terminal umbels of *Heracleum sosnowskyi* were sprayed with the phytohormone gibberellic acid (GA_3_) (SERVA, Heidelberg, Germany) at a 150 mg/L concentration, which was dissolved in distilled water. Certain order umbels were sprayed twice. Distilled water was used as a control spray. Manual sprayers “Venus” (KWAZAR, Budy-Grzybek, Poland) were used for spraying. Inflorescences were sprayed from a distance of 15–20 cm, evenly applying 18.5 mL onto each umbel.

### 4.3. Sample Harvesting

Samples for morphological analysis of *Heracleum sosnowskyi* flowers and mericarps were collected in August–September from terminal umbels. To study ovary development dynamics in terminal umbels, ovary samples were collected from the central and lateral parts of the terminal umbel at 0, 3, and 10 days after application (at the beginning of flowering). For biochemical and molecular studies, flowers of different developmental stages were collected, weighed, flash-frozen in liquid nitrogen, and stored in a low-temperature freezer (Skadi Green line, ES) at −80 °C until analysis.

### 4.4. Bioinformatic Analysis of Heracleum sosnowskyi GAoxs Subfamilies

In order to perform a comprehensive phylogenomic analysis of the GAox subfamilies in *Heracleum sosnowskyi*, a prerequisite was the in silico extraction of all potential GAox homolog sequences from publicly available whole-genome sequence data. We extracted 16 *Arabidopsis thaliana* and 21 rice GAox sequences from NCBI Phytozome and Rice Genome Annotation Project databases. These sequences were grouped according to protein subfamilies and used as query sequences against *H. sosnowskyi* genome (NCBI: PRJNA928505) in BLASTp [87]. Overall, 32 amino acid homologues were obtained and revised. Candidates were accepted if they shared at least 40% identity and had an expected threshold e-value ≤ 1.0 × 10^−10^. Subsequently, HsGAoxs protein candidates were submitted to InterPro Pfam database to identify the highly conserved and characteristic 2OG-FeII_Oxy (PF03171) and DIOX_N (PF14226) domains of the 2-ODDs superfamily [60,61]. Conserved motif analysis of HsGAoxs was performed using Multiple Em for Motif Elicitation (MEME) 5.5.7 software with default parameters [88]. The exon–intron gene structure of HsGAoxs genes was analyzed using GSDS 2.0 online software [89]. Based on the CDS region range found in the NCBI database annotation of each HsGAox protein, we visualized gene locations on *Heracleum sosnowskyi* chromosomes using MapGene2Chromosome v2 online tool [90]. In a subsequent analysis, the molecular weights of all HsGAox proteins were calculated using the Protein Molecular Weight tool from the www.bioinformatics.org website accessed on 10 March 2025. Additionally, the Plant-mPLoc 2.0 software was applied to predict the subcellular localizations of HsGAox proteins [91]. Protein structures of HsGAoxs were predicted using Alphafold 3 [92], and the predicted model structure quality was assessed by plDDT and PAE values, homology modeling and VoroMQA online tool using default parameters accessed on 10 March 2025 [93]. PAE and VoroMQA quality assessment are provided in Appendix A.

### 4.5. Multiple Sequence Alignment and Phylogenetic Analysis

Multiple sequence alignment of the GAoxs protein sequences from *Heracleum sosnowskyi*, *Arabidopsis thaliana*, and *Oryza sativa* was conducted using MEGA X software through the application of MUSCLE (Multiple Sequence Comparison by Log-Expectation) algorithm with default parameters [94].

An optimal model fit for aligned protein sequences was generated, and a phylogenetic tree was constructed using maximum likelihood (ML) and Le-Gascuel model by MEGA 12 software [95]. The robustness of the phylogenetic inference was evaluated using 500 bootstrap replicates.

### 4.6. Extraction of Endogenous GAs

Endogenous GAs were extracted from 1 g of fresh mericarp tissues that had been frozen in liquid nitrogen (N_2_) according to the specified methodology [22]. Each extract was analyzed with three biological replicates for each treatment. GA extraction consisted of three stages. The first stage entailed sample extraction with 80% methanol (purity ≥ 99.95%, Carl Roth GmbH + Co. KG, Karlsruhe, Germany), introduction of a deuterated 17,17-d_2_-GA standard mixture (internal standard) into the sample (purity ≥ 90%, OlChemim, Olomouc, Czech Republic), and liquid–liquid extraction. The second stage involved solid-phase extraction using anion exchange “Bond Elut DEA” (Agilent, Santa Clara, CA, USA) and reverse-phase “Sep-pak C18” (Waters, Wexford, Ireland) columns. The third stage was derivatization, during which the extracted samples were methylated with freshly prepared diazomethane according to De Boer and Backer [96] and trimethylsilylated with *N*-methyl-*N*-(trimethylsilyl)trifluoroacetamide (MSTFA) (purity ≥ 95%, Macherey-Nagel, Düren, Germany). After extraction, the samples were evaporated under a stream of N_2_ gas and stored at −20 °C until analysis by GC-MS method.

### 4.7. Endogenous GA Analysis by Gas Chromatography-Mass Spectrometry

Qualitative analysis of endogenous Gas was performed by GC-MS using a “GC/MS-Q2010 PLUS” (Shimadzu, Kyoto, Japan) gas chromatography system coupled with single quadrupole “GC-MS-QP2010 ULTRA” (Shimadzu, Kyoto, Japan) mass spectrometer. A total of 1 μL of the sample was manually injected into the injector of the gas chromatography system coupled to a non-polar “BPX5” column (30 m × 0.25 mm × 0.25 μm) (SGE Analytical Science, Kiln Farm Milton Keynes, UK). Helium was used as the carrier gas. The split ratio was 30:1 for sample introduction into the column. The initial column temperature was 60 °C and was immediately raised at a rate of 45 °C/min to 220 °C, followed by a gradual increase at 4 °C/min intervals up to 300 °C. The interface temperature was 240 °C. The analysis time for one sample was 25 min. Data were recorded in SIM (selected ion monitoring) mode 5 min after injection. Target ions for GA qualitative analysis were used for the corresponding GAs and d_2_-GA: 300 and 302 (GA_12_), 239 and 241 (GA_15_), 314 and 316 (GA_24_), 270 and 272 (GA_9_), 284 and 286 (GA_4_), 506 and 508 (GA_34_), 284 and 286 (GA_51_), 207 and 209 (GA_53_), 207 and 209 (GA_44_), 374 and 376 (GA_19_), 418 and 420 (GA_20_), 506 and 508 (GA_1_), 594 and 596 (GA_8_), and 506 and 508 (GA_29_). Compound identification was based on their retention times and the presence of additional characteristic ions in the obtained mass spectra, by comparing them with the known mass spectra of labeled and unlabeled GA methyl esters [97]. Endogenous GA levels were calculated based on peak areas, with corrections for the abundance of natural isotopes in the samples and the presence of unlabeled GAs in the internal GA standards [98].

### 4.8. Identification of GA Biosynthesis Gene Fragments

To amplify putative *Heracleum sosnowskyi GA 20-oxidase* (*GA20ox*), *GA 3-oxidase* (*GA3ox*), and *GA 2-oxidase* (*GA2ox*) gene fragments, a PCR-based method was applied [22,99]. Based on the conserved amino acid sequences and nucleotide sequences of the GA20ox and GA3ox enzyme subfamilies in wild carrot (*Daucus carota*) (Apiaceae), degenerate primers were designed to amplify these genes. For *GA2ox*, a degenerated primer pair from cucumber (*Cucumis sativus*) (Cucurbitacea) was used. cDNA molecules were synthesized from frozen fresh mericarp samples of *H. sosnowskyi*. The resulting cDNA was used as a template in a PCR reaction, which was performed using the “Phusion High-Fidelity PCR Master Mix” (Thermo Scientific, Vilnius, Lithuania) kit, according to the manufacturer’s protocol. PCR products were analyzed by electrophoresis on a 2% agarose gel, stained with “Midori Green Advance” (Nippon Genetics, Duren, Germany) dye and visualized with a “FastGeneR GelPic LED Box” device (Nippon Genetics, Duren, Germany). The expected size of the *DcGA20ox* PCR product was 813 base pairs (bp), the expected size of the *DcGA3ox* product was 497 bp, and the expected size of the *CsGA2ox* product was 300 bp. PCR product bands were cut from the agarose gel and samples were purified using the “Gene JET^TM^ PCR Purification Kit” according to the manufacturer’s instructions (Thermo Scientific, Vilnius, Lithuania). Samples were prepared for sequencing according to the recommendations of “Microsynth SEQLAB” (Göttingen, Germany). Sequencing results allowed for the identification of regions in the *H. sosnowskyi* genome based on which specific primer pairs were designed. Using “NCBI PrimerBLAST” accessed on 10 March 2025 and “Primer3” software version 4.1.0, forward (FW) and reverse (REV) primers were generated for the putative *H. sosnowskyi HsGA20ox1*, *HsGA3ox1*, and *HsGA2ox1* genes (Appendix A).

### 4.9. Real-Time Quantitative PCR

Real-time quantitative PCR (qPCR) was performed to assess the transcript levels of late gibberellin biosynthesis enzyme (GA20ox, GA3ox, GA2ox) genes in *Heracleum sosnowskyi* ovary samples. Specific primers were used in the qPCR analysis to evaluate the expression of the genes encoding these proteins (Appendix A). An amount of 1 µL of cDNA diluted 5 times with double distilled water was used as the template for the qPCR reaction. The total reaction mixture volume was 15 µL. Reactions were performed in an “AZURE CIELO^TM^ Real-time PCR System” (Azure Biosystems, Dublin, TX, USA) thermocycler using the two-step cycle protocol of the “Maxima SYBR Green noROX qPCR Master Mix (2X)” (Thermo Scientific, Vilnius, Lithuania) kit. Cycling conditions consisted of one cycle at 95 °C for 10 min and 40 cycles at 95 °C for 15 s, followed by 60 °C for 1 min. The cucumber (*Cucumis sativus*) β-actin gene (AB010922) was used as the internal standard. The relative abundance of each gene was calculated as the average of two technical replicates using the “Real-time PCR Miner” bioinformatics software accessed on 10 March 2025 and expressed as a percentage [100]. Technical replicates with a Ct difference ≥0.5 from the mean were discarded. All experiments were repeated twice with three biological replicates.

### 4.10. Statistical Analysis

Using descriptive statistical methods, the obtained results were presented as arithmetic means with standard errors (mean ± SE). The normal distribution of data was assessed using the Shapiro–Wilk test. If the assumption of data normality was met, parametric methods were employed, and the homoscedasticity of the data was assessed using Levene’s test with Brown–Forsythe correction. If the data were not normally distributed, statistically significant differences between group medians were assessed using the Kruskal–Wallis H-test. For pairwise comparisons between experimental groups, the Mann–Whitney U test or Dunn’s z test was applied. Statistically significant differences were indicated when *p* < 0.05. Data statistical analysis and graphical representation were performed using “PAST 4.16” and “R” version 4.0.2 in “RStudio” version 1.3.1093 with “ggplot2” package version 3.5.2 [101,102].

## 5. Conclusions

A total of 27 *HsGAox* genes were widely distributed across eleven *Heracleum sosnowskyi* chromosomes. Phylogenetic classification allows us to make speculations about the functions of the HsGA20ox, HsGA2ox, and HsGA3ox. The analyzed *H. sosnowskyi* proteins clustered into three different subfamilies. Among thirteen HsGA2ox members, none belonged to the C_20_-GA2ox subfamily, contrary to the two subfamilies of GA2oxs in other plant species. Gene structure and motif analyses support the phylogenetic findings, indicating the conserved evolution of these gene subfamilies across three species. Intense fruit development is dependent on endogenous GA, as supported by the obtained levels of endogenous GA. The phenotypic and morphometric analysis of ovaries corresponds to the peak of gibberellin during the cell expansion phase. However, the effect of exogenous GA_3_ stimulated *HsGA3ox1* expression in the central and lateral parts of the umbel ovaries. Overall, these results confirm that GA homeostasis is a tightly regulated, complex system including multiple *HsGAox* genes. It maintains a stable hormone level in the early stages of fruit development. A high or low level of the hormone in a particular umbel part can disrupt the development of the fruit. These results open opportunities to further analyze the role of GAs in *H. sosnowskyi* fruit set mechanism and develop invasion control strategies.

## Figures and Tables

**Figure 1 ijms-26-04480-f001:**
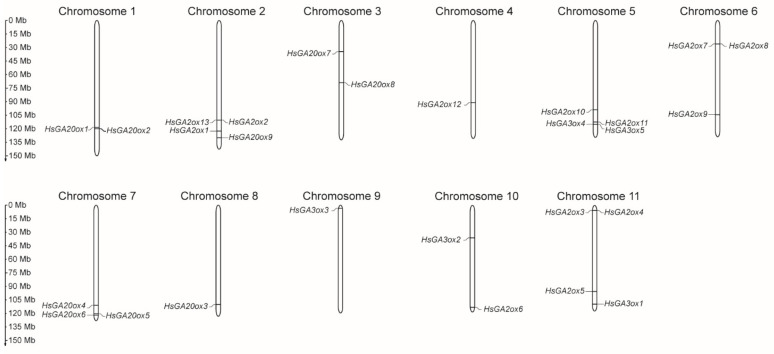
Distribution of *HsGAox* genes on *Heracleum sosnowskyi* chromosomes. The scale refers to the lengths of the chromosomes.

**Figure 2 ijms-26-04480-f002:**
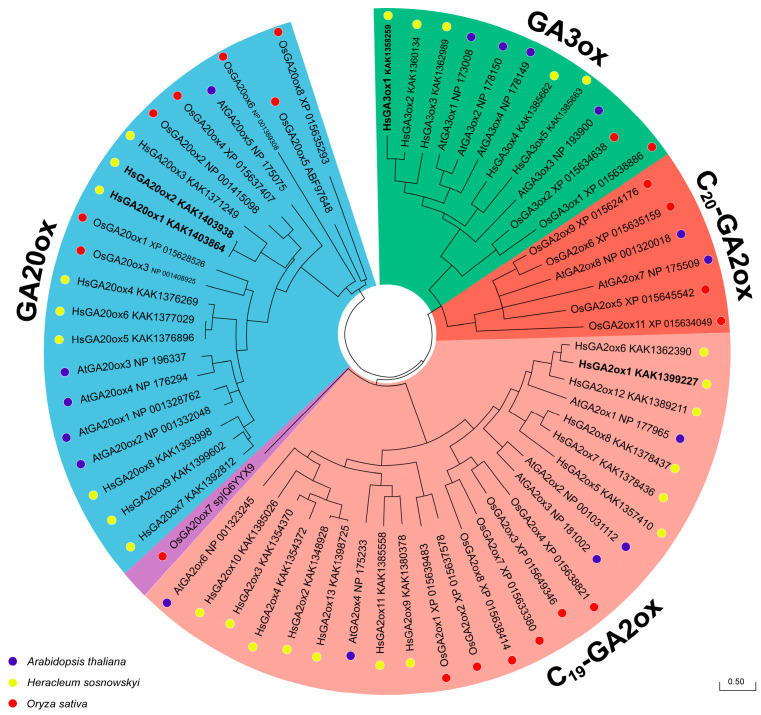
The phylogenetic relationship analysis of GAox proteins among *Heracleum sosnowskyi*, *Arabidopsis thaliana*, and *Oryza sativa*. The tree was constructed according to the ML method, using MEGA 12.0 software with 500 bootstrap replicates. The GAox subfamilies and analyzed species were labeled with different colors. Scale bar refers to a phylogenetic distance of amino acid substitutions per site.

**Figure 3 ijms-26-04480-f003:**
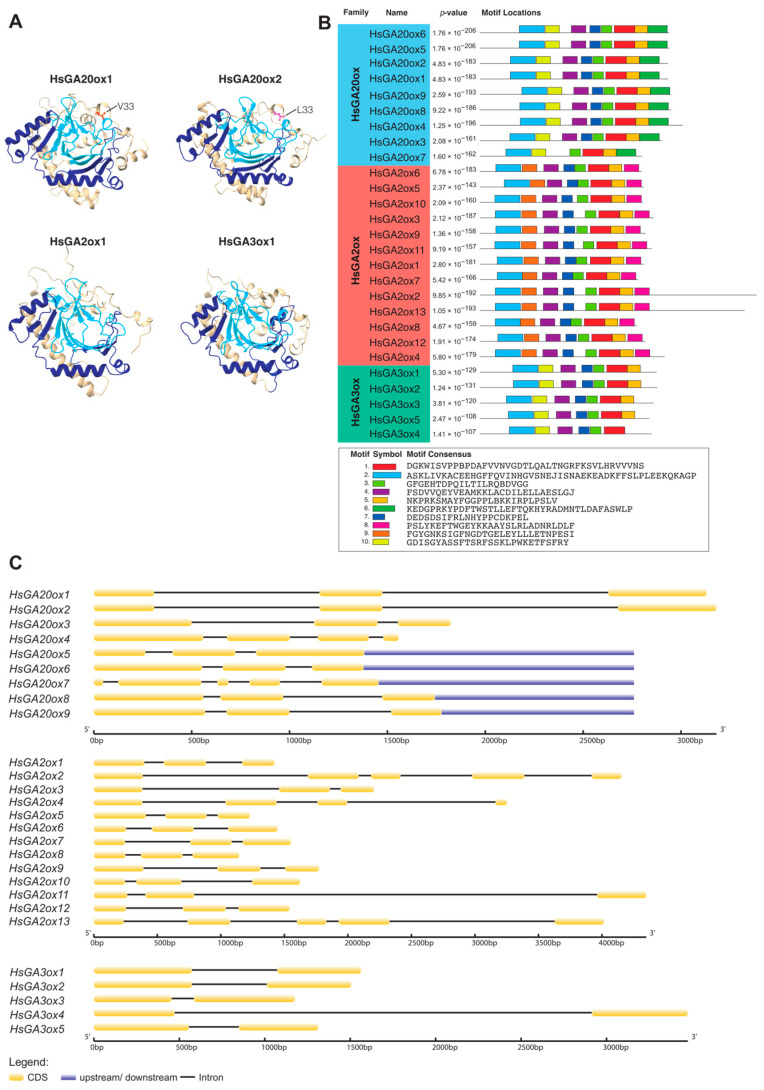
Representation of conserved protein domains, motifs, and gene structure patterns of *Heracleum sosnowskyi* GAox. (**A**) Generated models of putative HsGA20ox1, HsGA20ox2, HsGA2ox1, and HsGA3ox1 proteins. Non-haem dioxygenase N-terminal and 2OG-Fe(II) oxygenase domains are highlighted in dark blue and light blue, respectively. (**B**) Schematic view of HsGAox protein sequences and 10 conserved motifs distributed among them. (**C**) The gene structure map of HsGAox members.

**Figure 4 ijms-26-04480-f004:**
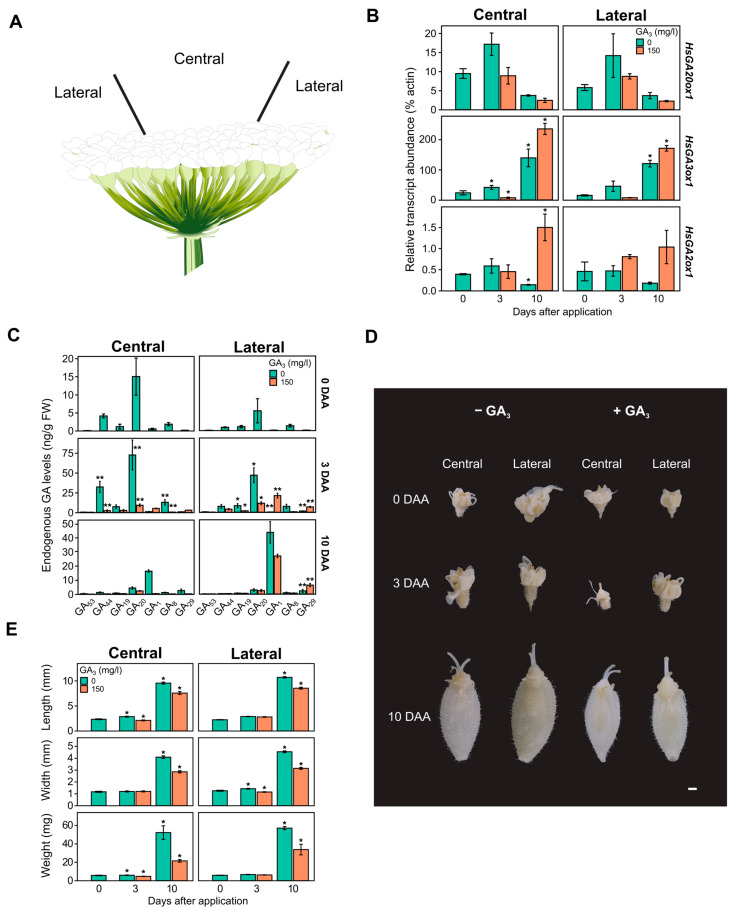
Schematic picture of *Heracleum sosnowskyi* terminal inflorescence (**A**). The effect of 150 mg/L gibberellic acid (GA_3_) on GA biosynthesis gene expression (**B**), endogenous gibberellin (GA) content (**C**), phenotype (**D**), and morphometry (**E**) in *H. sosnowskyi* ovary tissues from the central and lateral parts of terminal inflorescence. DAA—days after application GA_3_. Error bars represent the standard error of the mean. Asterisks in the same color columns indicate statistically significant differences (*—*p* < 0.01; **—*p* < 0.001). Scale bar represents 1 mm.

**Table 1 ijms-26-04480-t001:** Characterization of the HsGAox superfamily in *Heracleum sosnowskyi*.

Protein Name	Accession no.	Gene Name	CDS (nt)	Length (aa)	MW (kDa)	Domain Location	Type	Subcellular Localization
HsGA20ox1	KAK1403938	*HsGA20ox1*	1134	377	42.88	44–144 ^a^ 209–306 ^b^	GA20ox	Cytoplasm
HsGA20ox2	KAK1403864	*HsGA20ox2*	1134	377	42.89	44–144 ^a^ 209–306 ^b^	GA20ox	Cytoplasm
HsGA20ox3	KAK1371249	*HsGA20ox3*	1095	364	41.33	43–144 ^a^ 209–306 ^b^	GA20ox	Cytoplasm
HsGA20ox4	KAK1376269	*HsGA20ox4*	1221	406	46.14	63–169 ^a^ 227–325 ^b^	GA20ox	Cytoplasm
HsGA20ox5	KAK1376896	*HsGA20ox5*	1140	379	42.96	61–167 ^a^ 225–323 ^b^	GA20ox	Cytoplasm
HsGA20ox6	KAK1377029	*HsGA20ox6*	1140	379	43.02	61–167 ^a^ 225–323 ^b^	GA20ox	Cytoplasm
HsGA20ox7	KAK1392812	*HsGA20ox7*	978	325	36.81	62–164 ^a^ 227–325 ^b^	GA20ox	Cytoplasm
HsGA20ox8	KAK1393998	*HsGA20ox8*	1152	383	43.73	34–135 ^a^ 181–260 ^b^	GA20ox	Cytoplasm
HsGA20ox9	KAK1399602	*HsGA20ox9*	1149	382	43.38	66–170 ^a^ 230–328 ^b^	GA20ox	Cytoplasm
HsGA2ox1	KAK1399227	*HsGA2ox1*	987	328	36.87	26–115 ^a^ 176–273 ^b^	C_19_-GA2ox	Cytoplasm
HsGA2ox2	KAK1348928	*HsGA2ox2*	1665	554	61.92	20–79 ^a^ 172–291 ^b^	C_19_-GA2ox	Cytoplasm Nucleus
HsGA2ox3	KAK1354370	*HsGA2ox3*	1047	348	38.86	20–88 ^a^ 173–291 ^b^	C_19_-GA2ox	Cytoplasm
HsGA2ox4	KAK1354372	*HsGA2ox4*	1113	370	41.05	20–88 ^a^ 173–292 ^b^	C_19_-GA2ox	Cytoplasm
HsGA2ox5	KAK1357410	*HsGA2ox5*	987	328	36.77	40–96 ^a^ 180–275 ^b^	C_19_-GA2ox	Cytoplasm
HsGA2ox6	KAK1362390	*HsGA2ox6*	975	342	36.31	23–110 ^a^ 173–269 ^b^	C_19_-GA2ox	Cytoplasm
HsGA2ox7	KAK1378436	*HsGA2ox7*	954	317	35.81	26–111 ^a^ 170–265 ^b^	C_19_-GA2ox	Cytoplasm
HsGA2ox8	KAK1378437	*HsGA2ox8*	945	314	35.39	22–123 ^a^ 166–261 ^b^	C_19_-GA2ox	Cytoplasm
HsGA2ox9	KAK1380378	*HsGA2ox9*	999	332	37.21	21–80 ^a^ 174–275 ^b^	C_19_-GA2ox	Cytoplasm
HsGA2ox10	KAK1385026	*HsGA2ox10*	981	326	36.65	19–88 ^a^ 169–276 ^b^	C_19_-GA2ox	Cytoplasm
HsGA2ox11	KAK1385558	*HsGA2ox11*	1035	344	38.19	19–84 ^a^ 172–288 ^b^	C_19_-GA2ox	Cytoplasm
HsGA2ox12	KAK1389211	*HsGA2ox12*	996	331	37.05	27–118 ^a^ 177–276 ^b^	C_19_-GA2ox	Cytoplasm
HsGA2ox13	KAK1398725	*HsGA2ox13*	1593	530	59.09	20–79 ^a^ 172–291 ^b^	C_19_-GA2ox	Cytoplasm
HsGA3ox1	KAK1358259	*HsGA3ox1*	1065	354	39.98	56–158 ^a^ 211–308 ^b^	GA3ox	Cytoplasm
HsGA3ox2	KAK1360134	*HsGA3ox2*	1068	355	40.02	57–159 ^a^ 210–309 ^b^	GA3ox	Cytoplasm
HsGA3ox3	KAK1362989	*HsGA3ox3*	1047	348	39.05	46–145 ^a^ 196–295 ^b^	GA3ox	Cytoplasm
HsGA3ox4	KAK1385662	*HsGA3ox4*	1035	344	38.59	48–141 ^a^ 201–297 ^b^	GA3ox	Cytoplasm
HsGA3ox5	KAK1385663	*HsGA3ox5*	1020	339	38.00	50–152 ^a^ 206–302 ^b^	GA3ox	Cytoplasm

aa: amino acid; CDS: coding sequence; MW: the theoretical molecular weight of proteins; nt: nucleotide; ^a^: non-haem dioxygenase N-terminal domain; ^b^: 2OG-Fe(II) oxygenase domain.

## Data Availability

The data supporting the reported results can be found in the archive of scientific reports of the Nature Research Centre.

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
