# Peer review of "HsGA20ox1*, *HsGA3ox1*, and *HsGA2ox1* Are Involved in Endogenous Gibberellin Regulation Within *Heracleum sosnowskyi* Ovaries After Gibberellin A_3_ Treatment"

_ijms, 2025, doi:10.3390/ijms26104480_

Round 1

Reviewer 1 Report

Comments and Suggestions for Authors

The Heracleum sosnowskyi is a noxious invasive plant, HsGA20ox1, HsGA3ox1 and HsGA2ox1 of family genes in H. sosnowskyi were identified by bioinformatics methods, also endogenous GAs levels and expression of some HsGAoxs were analyzed. This study gave us a large amount of valuable information of gibberellin metabolism in H. sosnowskyi, but there are some place which could be improved. Some comments are as follows,

  1. in the method, how to dissolve the GAshould be introdued
  2. how much fresh mericarp tissues be used to extraction of endogenous GAs, and repeated times of analysis
  3. only three genes of expression levels are estimulated, why selected these three HsGA20ox1, HsGA3ox1, HsGA2ox1
  4. The order of figures in Figure 4 need to be adjusted according to the sequence of the presentation

Author Response

Response to Reviewer 1 comments:

Point 1: In the method, how to dissolve the GAshould be introduced

Response 1: Thank you very much for the comment. In lines 373-374 the phrase„ which dissolved in distilled water“ was inserted.

Point 2: How much fresh mericarp tissues be used to extraction of endogenous GAs, and repeated times of analysis

Response 2: Thank you very much for the comment. In line 420 we added „1 gram of fresh mericarp tissues“ to detail the amount of material used for endogenous GAs extraction. In lines 421-422 was described times of repetition for analysis including: „Each extract had been analysed with three biological replicates for each treatment“.

Point 3: Only three genes of expression levels are estimulated, why selected these three HsGA20ox1, HsGA3ox1, HsGA2ox1

Response 3: Thank you for the question. Using degenerated primer pairs, we identified HsGA20ox1, HsGA3ox1 and HsGA2ox1 as the most plausible abundant GAox genes within Heracleum sosnowskyi early development ovaries and used them in further analysis.

Point 4: The order of figures in Figure 4 need to be adjusted according to the sequence of the presentation

Response 4: Thank you very much for the comment. Elements in Figure 4 were rearranged according to the main text's sequence.

Reviewer 2 Report

Comments and Suggestions for Authors

Review of the article "HsGA20ox1, HsGA3ox1 and HsGA2ox1 are Involved in Endogenous Gibberellin Regulation within Heracleum sosnowskyi Ovaries after GA3 Treatment"

I think the paper is important and relevant, well structured. It combines  bioinformatic analyses with experimental material. The study focuses on endogenous gibberellin concentrations and the analysis of genes related to its accumulation in the ovaries of the invasive plant Heracleum sosnowskyi during fruit development. The authors identified the effects of exogenous GA3 on the expression of HsGA20ox1, HsGA3ox1 and HsGA2ox1, as well as on the endogenous GA profiles in gradually opening H. sosnowskyi flowers. The results of the paper open opportunities for further developm of invasion control strategies.

I have a few comments on the manuscript:

Throughout the paper, the names of plant species should be spelled with the full genus name the first time it is used, and the full genus name is used again when the name of another plant is mentioned. The full manuscript should be reviewed.

Figure 4. In the title of the figure, the reference (A) should be moved after the first sentence.

Conclusions should avoid repetition of thought and emphasise the potential application of the study to the development of new technologies.

Author Response

Response to Reviewer 2 comments:

Point 1: Throughout the paper, the names of plant species should be spelled with the full genus name the first time it is used, and the full genus name is used again when the name of another plant is mentioned. The full manuscript should be reviewed.

Response 1: Thank you very much for the valuable comment. We agree with this comment. Full manuscript revised and corrected.

Point 2: Figure 4. In the title of the figure, the reference (A) should be moved after the first sentence.

Response 2: Thank you very much for the correction. The caption of Figure 4 was revised and corrected.

Point 3: Conclusions should avoid repetition of thought and emphasise the potential application of the study to the development of new technologies.

Response 3: Thank you very much for the comment. We rewrote the conclusions part according to your suggestion.
